# Harnessing the IL-21-BATF Pathway in the CD8^+^ T Cell Anti-Tumor Response

**DOI:** 10.3390/cancers13061263

**Published:** 2021-03-12

**Authors:** Paytsar Topchyan, Gang Xin, Yao Chen, Shikan Zheng, Robert Burns, Jian Shen, Moujtaba Y. Kasmani, Matthew Kudek, Na Yang, Weiguo Cui

**Affiliations:** 1Blood Research Institute, Versiti Wisconsin, Milwaukee, WI 53226, USA; ptopchyan@mcw.edu (P.T.); gang.xin@osumc.edu (G.X.); yao.chen@versiti.org (Y.C.); szheng@versiti.org (S.Z.); rtburns@versiti.org (R.B.); jshen@mcw.edu (J.S.); mkasmani@mcw.edu (M.Y.K.); mkudek@mcw.edu (M.K.); yn0198@163.com (N.Y.); 2Department of Microbiology and Immunology, Medical College of Wisconsin, Milwaukee, WI 53226, USA; 3Department of Microbial Infection and Immunity, The Ohio State University College of Medicine, Columbus, OH 43210, USA; 4Pelotonia Institute for Immuno-Oncology, The James Comprehensive Cancer Center, The Ohio State University, Columbus, OH 43210, USA; 5Department of Pediatrics, Medical College of Wisconsin, Milwaukee, WI 53226, USA; 6Department of Neurology, Kailuan General Hospital, Tangshan 063000, China

**Keywords:** BATF, IL-21, CD8 T cells, CD4 T cells, adoptive cell therapy, cancer immunotherapy

## Abstract

**Simple Summary:**

In cancer, CD8^+^ T cells enter a state of dysfunction within the tumor that prevents them from targeting and killing tumor cells. Our study aims to uncover how CD8^+^ T cells can be helped by CD4^+^ T cells or modified in order to improve their effector function against cancer. Thus, allowing them to better fight and control tumors. Our work shows that the protein Basic Leucine Zipper ATF-Like Transcription Factor (BATF) may be a key regulator of CD8^+^ T cells and their anti-tumor function. These findings can provide further insight for the development of novel therapeutic treatments for cancer patients.

**Abstract:**

In cancer, CD8^+^ T cells enter a dysfunctional state which prevents them from effectively targeting and killing tumor cells. Tumor-infiltrating CD8^+^ T cells consist of a heterogeneous population of memory-like progenitor, effector, and terminally exhausted cells that exhibit differing functional and self-renewal capacities. Our recently published work has shown that interleukin (IL)-21-producing CD4^+^ T cells help to generate effector CD8^+^ T cells within the tumor, which results in enhanced tumor control. However, the molecular mechanisms by which CD4^+^ helper T cells regulate the differentiation of effector CD8^+^ T cells are not well understood. In this study, we found that Basic Leucine Zipper ATF-Like Transcription Factor (BATF), a transcription factor downstream of IL-21 signaling, is critical to maintain CD8^+^ T cell effector function within the tumor. Using mixed bone marrow chimeras, we demonstrated that CD8^+^ T cell-specific deletion of BATF resulted in impaired tumor control. In contrast, overexpressing BATF in CD8^+^ T cells enhanced effector function and resulted in improved tumor control, bypassing the need for CD4^+^ helper T cells. Transcriptomic analyses revealed that BATF-overexpressing CD8^+^ T cells had increased expression of costimulatory receptors, effector molecules, and transcriptional regulators, which may contribute to their enhanced activation and effector function. Taken together, our study unravels a previously unappreciated CD4^+^ T cell-derived IL-21–BATF axis that could provide therapeutic insights to enhance effector CD8^+^ T cell function to fight cancer.

## 1. Introduction

Adoptive cell transfer immunotherapy has made great advancements towards effective treatments, especially for patients with hematological malignancies, but immunotherapeutic efficacy in treating those with solid tumors remains limited. A key factor for this setback in treating solid tumors is that tumor-infiltrating lymphocytes (TILs) often differentiate into dysfunctional states, resembling exhausted T cells that arise in chronic viral infections [1,2,3]. Along with the upregulation of inhibitory molecules, such as programmed cell death protein 1 (PD-1), the dysfunctional state of exhaustion in CD8^+^ T cells is characterized by diminished effector function, namely decreased cytotoxic activity and reduced expression of effector molecules such as granzyme B and interferon-γ (IFNγ) [1,4].

In prolonged inflammatory states such as chronic infection and cancer, CD8^+^ T cells are comprised of a heterogeneous population with differing functional characteristics, such as stem-like proliferative capacity and cytolytic effector function [5,6,7,8,9]. Interestingly, we and others have identified a progenitor or memory-like CD8^+^ T cell subset (TCF1^hi^ Ly108^+^), present in chronic infection and cancer, with a unique capacity for self-renewal [5,6,10,11]. Our recently published findings have demonstrated that these progenitor cells can give rise to two distinct CD8^+^ T cell subsets during chronic viral infection: CX_3_CR1^+^ cytolytic effector CD8^+^ T cells and terminally exhausted Ly108^−^CX_3_CR1^−^ PD-1^hi^ cells [5]. The effector CX_3_CR1^+^ subset exhibits augmented cytolytic function and secretion of effector molecules, such as granzyme B and IFNγ [10,11]. Consistently, CX_3_CR1 upregulation in CD8^+^ T cells is associated with positive outcomes in melanoma patients [12]. Alternatively, chronic antigen stimulation can cause progenitor CD8^+^ T cells to differentiate towards an exhausted state, counterproductive to overcoming viral infections or preventing tumor growth [13]. Thus, redirecting progenitor cell differentiation towards CX_3_CR1^+^ effector cells will likely help overcome functional exhaustion of tumor-reactive CD8^+^ T cells.

CD4^+^ T cells provide “help” to cytotoxic CD8^+^ T cells to support tumor control [14,15]. In chronic viral infection, CD4^+^ T cells are also necessary for sustained CD8^+^ T cell function [16,17]. In particular, studies have found a critical role for interleukin (IL)-21, a cytokine produced primarily by CD4^+^ T cells, in facilitating a CD8^+^ T cell response in both cancer and chronic infection [18,19,20,21]. More recently, our findings suggest that IL-21-producing CD4^+^ T cells are required for the formation of effector CX_3_CR1^+^ CD8^+^ TILs, which enhances tumor control [5]. However, the precise molecular mechanisms by which IL-21-producing CD4^+^ T cells coordinate this anti-tumor effect are not well understood.

We have previously demonstrated that IL-21 signaling through signal transducer and activator of transcription 3 (STAT3) induces Basic Leucine Zipper ATF-Like Transcription Factor (BATF) activation in CD8^+^ T cells, resulting in their sustained survival and effector function during chronic viral infection [21]. BATF is a pioneer transcription factor (TF) known to facilitate changes in the chromatin landscape, promoting the differentiation of effector CD8^+^ T cells in the chronic viral infection model [22]. However, its role in cancer is not well studied. We, therefore, sought to assess the CD8^+^ T cell-intrinsic requirement of BATF downstream of IL-21 signaling, its role in effector CD8^+^ T cell differentiation and function in cancer, and its utility as a potential immunotherapeutic target.

## 2. Results

### 2.1. BATF Is Intrinsically Required for CD8^+^ T Cell Effector Function within the Tumor

To determine if BATF is intrinsically required for CD8^+^ T cell-mediated tumor control, we utilized a mixed bone marrow chimera (BMC) model. To do this, *Cd8a*^−/−^ recipient mice were reconstituted with *Cd8a*^−/−^ (70%) and either wild-type (*Batf^+/+^)* or *Batf*^−/−^ (30%) bone marrow. Normal CD4 and CD8 T cell development is observed in *Batf*^−/−^ mice [23], supporting the use of this mixed BMC model to study CD8 T cells’ differentiation in melanoma. Subsequently, we inoculated mixed BMC mice with B16-F10 melanoma and monitored their growth and survival. Mice with *Batf*^−/−^ CD8^+^ T cells had significantly larger tumors compared to wild-type controls (Figure 1A). The BATF-deficient group also exhibited reduced survival (Figure 1B). These findings indicate that BATF is intrinsically required for CD8^+^ T cell-mediated tumor control.

Next, we utilized this mixed BMC model to assess tumor-infiltrating CD8^+^ T cell phenotypes and functions. We initially inoculated mixed BMC mice with B16-GP33 tumors in order to assess GP33 tetramer-specific CD8^+^ T cells. However, too few tetramer-specific cells were detected (data not shown); therefore, total activated (CD44^+^) CD8^+^ T cells were instead analyzed for all future experiments. There was a significantly lower number of tumor-infiltrating CD8^+^ T cells per mm^3^ of tumor volume from the *Batf^−/−^* group (Figure 1C). Additionally, BATF-deficient tumor-reactive CD8^+^ T cells had significantly lower expression of granzyme B, a cytolytic molecule produced by effector lymphocytes (Figure 1D). In summary, these findings suggest that BATF is an essential component of CD8^+^ T cell anti-tumor effector function.

### 2.2. BATF Acts Downstream of IL-21 Signaling to Enhance the Anti-Tumor CD8^+^ T Cell Response

Adoptive cell transfer (ACT) of CD4^+^ helper T cells has shown promise in cancer immunotherapy [24,25]. Interestingly, other studies have found that IL-21 may be responsible, in part, for the anti-tumor functions of helper CD4^+^ T cells [26]. Our group has previously shown that IL-21-producing CD4^+^ T cells provide critical help to promote the generation of effector CX_3_CR1^+^ CD8^+^ T cells in a preclinical melanoma model [5]. Therefore, we next tested whether BATF, which is downstream of IL-21 signaling, is necessary for endogenous CD8^+^ T cell effector differentiation in response to ACT of IL-21-producing CD4^+^ T cells. Utilizing our mixed BMC model (Figure 1), melanoma tumor-bearing mice were adoptively transferred with IL-21-producing CD4^+^ T cells. Prior to ACT, tumor-specific CD4^+^ T cells were cultured in Th17 skewing conditions to produce IL-21, as we have previously published [5,27]. Tumor volumes were measured immediately before ACT and at multiple time points post-ACT (Figure 2A). Eight days following treatment, harvested tumors were assessed for immune cell phenotype and function. *Batf^−/−^* mice exhibited significantly lower numbers of tumor-infiltrating CD8^+^ T cells per mm^3^ of tumor volume as compared to wild-type controls following IL-21-producing CD4^+^ T cell ACT (Figure 2B). Upon assessment of tumor-reactive CD8^+^ T cells, we found that the *Batf^−/−^* group had a significantly lower frequency of effector CX_3_CR1^+^ CD8^+^ T cells with a concomitant increase in Ly108^−^ CX_3_CR1^−^ exhausted CD8^+^ T cells (Figure 2C). Furthermore, tumor-infiltrating *Batf^−/−^* CD8^+^ T cells had significantly lower expression of the cytolytic effector molecule granzyme B (Figure 2D). Interestingly, surface expression of PD-1 was significantly lower in BATF-deficient CD8^+^ T cells than their wild-type counterparts (Figure 2E). PD-1, although generally regarded as an inhibitory receptor, is also indicative of T cell activation [28,29], suggesting that BATF plays an important role in the maintenance of activated CD8^+^ T cells in the tumor. These findings indicate that BATF is indispensable in effector CX_3_CR1^+^ CD8^+^ T cell anti-tumor function in response to IL-21-producing CD4^+^ T cell immunotherapy. 

### 2.3. BATF Overexpression in Tumor-Specific CD8^+^ T Cells Enhances Tumor Infiltration and Effector Function, Bypassing the Need for CD4^+^ T Cell Help

After observing the critical role of BATF downstream of IL-21 signaling in tumor-infiltrating effector CD8^+^ T cell differentiation, infiltration, and function, we wanted to explore whether BATF overexpression in tumor-specific CD8^+^ T cells could bypass their need for CD4^+^ T cell help. We first retrovirally transduced Pmel CD8^+^ T cells with either MSCV-IRES-Thy1.1 (MIT)-Empty vector or MIT-BATF retrovirus (RV) (Appendix A). CD8^+^ T cells from Pmel mice express transgenic T cell receptors specific to the gp100 antigen of B16 melanoma, making them a perfect candidate for tumor-specific CD8^+^ T cells for our adoptive cell therapy treatment. MIT-BATF-transduced CD8^+^ T cells exhibited higher expression of granzyme B than CD8^+^ T cells transduced with MIT-Empty vector (Figure 3A, top). Furthermore, BATF-overexpressing cells showed reduced expression of the transcription factor T cell factor 1 (TCF1) (Figure 3A, bottom), indicating enhanced effector CD8^+^ T cell differentiation, as downregulated TCF1 is also observed in progenitor-to-effector CD8^+^ T cell transition during chronic lymphocytic choriomeningitis virus LCMV infection [6,7]. Mice treated with BATF-overexpressing CD8^+^ T cells maintained significantly slower tumor growth compared to those that received the empty vector-transduced CD8^+^ T cell treatment (Figure 3B). One week following treatment, tumors were harvested for assessment. Tumors of mice treated with MIT-BATF had significantly more tumor-infiltrating, adoptively transferred (Thy1.1^+^) CD8^+^ T cells compared to MIT-Empty vector control mice (Figure 3C). Further assessment of these adoptively transferred cells indicated that BATF overexpression enhances CX_3_CR1 surface expression (Figure 3D) and production of granzyme B (Figure 3E). Thus, these findings suggest that BATF is a critical transcription factor that, when overexpressed, can enhance the tumor infiltration, survival, and function of tumor-specific cytotoxic effector CD8^+^ T cells, resulting in significantly reduced tumor growth.

### 2.4. BATF-Overexpressing Tumor-Specific CD8^+^ T Cells Exhibit an Effector Program Similar to CD8^+^ T Cells Found in LCMV Infection

As our previous findings indicated, BATF overexpression enhanced tumor infiltration and function of effector CD8^+^ T cells within the tumors of mice treated with this adoptive cell therapy. To better understand how BATF may regulate this enhanced effector program, we conducted bulk RNA sequencing of tumor-infiltrating, adoptively transferred transduced CD8^+^ T cells. Principle component analysis showed that tumor-reactive BATF-overexpressing CD8^+^ T cells exhibited a distinct transcriptional profile compared to control empty vector-transduced CD8^+^ T cells (Appendix A). To assess the changes in these tumor-reactive CD8^+^ T cells upon BATF overexpression, we conducted gene set enrichment analysis (GSEA) using gene sets from the Broad Institute’s Molecular Signatures Database (MSigDB) and the Reactome database. Analysis of immunologic gene signature and hallmark gene sets revealed that BATF-overexpressing tumor-specific CD8^+^ T cells found in the tumor exhibited a transcriptional profile consistent with higher effector, higher memory, and lower exhausted gene expression profiles (Figure 4A). Our findings also indicated that, compared to empty vector controls, tumor-specific CD8^+^ T cells that overexpressed BATF expressed a gene set more similar to CD8^+^ T cells found in acute, rather than chronic, LCMV infection (Figure 4A,B). Conversely, relative to BATF-overexpressing cells, empty vector control cells upregulated a second set of genes enriched in chronic LCMV infection compared to acute LCMV infection (Appendix A). These associations with T cell effector functions were also observed when benchmarking our cells against gene signatures of multiple CD8^+^ T cell phenotypes from chronic LCMV infection. Here, relative to empty vector-transduced control CD8^+^ T cells, BATF-overexpressing CD8^+^ T cells exhibited enrichment for a gene set upregulated in early effector CD8^+^ T cells compared to late exhausted CD8^+^ T cells (Figure 4C). These data collectively suggest that BATF overexpression results in a more functional CD8^+^ T cell effector profile, rather than an exhausted profile.

Taking a closer look at some genes present in these gene sets, as well as other key effector CD8^+^ T cell markers, we found that BATF-overexpressing, tumor-infiltrating CD8^+^ T cells significantly upregulated genes encoding activation markers (*Cd44*, *Ifngr1*, and *Rara*), costimulatory molecules (*Cd28* and *Icos*), effector molecules (*Gzma*, *Gzmb*, *Gzmc*, and *Gzmf*), chemokine receptors (*Cxcr3*, *Cxcr4*, *Cxcr6*, *Ccr2*, and *Ccr5*), and transcription factors (*Id2*, *Zbtb9*, *Smad3*, *Lef1*, and *Hif1a*) (Figure 4D), many of which are known target genes of BATF [22]. Upregulation of granzymes, in particular granzyme B, is a distinguishing marker of T cell effector function. Furthermore, when analyzing Reactome pathways, we found that the FOXO-mediated transcriptional program, which regulates cell survival, differentiation, cell growth, and metabolism during environmental stresses [30,31,32], was highly enriched in BATF-overexpressing CD8^+^ T cells (Appendix A). Interestingly, the transcription factor *Hif1a*, which is important in mediating the cellular response to hypoxic environments, was more highly expressed in BATF-overexpressing CD8^+^ T cells (Figure 4D), indicating that these cells may be better adapted to the hypoxic tumor microenvironment, as supported by previous work [33]. Additionally, GSEA revealed that BATF-overexpressing CD8^+^ T cells were negatively enriched for Reactome pathways of cellular responses to stress as well as stress-induced senescence (Appendix A). Furthermore, certain markers of epigenetic regulation (*Tox*, *Hat1*, and *Dnmt3a*) were downregulated in BATF-overexpressing CD8^+^ T cells (Figure 4D), along with negative regulators of rRNA expression (Appendix A). Among these, the transcription factor *Tox* is of specific interest as it is known to be upregulated in exhausted T cells in the settings of chronic infection and cancer [34,35,36]. Collectively, these findings support the idea that BATF is a key regulator of effector CD8^+^ T cell activity and function within the tumor and shed light on potential pathways that BATF may upregulate to facilitate effective tumor control.

## 3. Discussion

In this study, we established the critical role of the IL-21–BATF pathway in maintaining functional tumor-reactive CD8^+^ T cells in cancer. CD8^+^ T cells require IL-21-producing CD4^+^ T cells’ help to undergo differentiation towards a functional effector state [5]; we have now shown that BATF is intrinsically necessary for this process. We have additionally shown that by overexpressing BATF in CD8^+^ T cells, we can enhance CD8^+^ T cell infiltration, effector differentiation, and function in melanoma. Finally, transcriptional assessment of adoptively transferred BATF-overexpressing CD8^+^ T cells revealed a more effector-like and less exhausted program according to several GSEA and gene-level differential expression analyses. These results reveal that BATF overexpression may bypass the need for CD4^+^ T cell help to prevent exhaustion and enhance effector function, providing an avenue for cellular anti-tumor therapeutic intervention.

CD4^+^ T cells are not only important for providing help to CD8^+^ T cells via antigen presentation, but also help in facilitating a cytotoxic T cell effector differentiation program [37,38]. Unfortunately, the tumor microenvironment (TME) is largely immunosuppressive, with the predominant CD4^+^ T cell population found in tumors being regulatory T cells [15,39]. Interestingly, as previously observed in the B16 melanoma model, CD8^+^ T cells primarily exist in two dysfunctional states within the TME, either terminally exhausted or progenitor-like [40], which resemble an immunological state similar to “unhelped” CD8^+^ T cells in the setting of chronic LCMV infection [5,41]. Our recently published work has shown that when CD4^+^ T cell help is removed via CD4 depletion, the development of CX_3_CR1^+^ effector CD8^+^ T cells in chronic LCMV infection is abrogated [41], thus mimicking the CD4^+^ T cell-deplete TME. Conversely, ACT of CD4^+^ helper T cells has shown promise in cancer immunotherapy models [24,25], where they function as a “living drug” via cytokine production. Most notably, Th17 and Th9 cells appear to be a more effective than Th1 cells in limiting tumor progression [42]. Although the precise mechanisms by which they mediate this anti-tumor effect are not well understood, IL-21, commonly produced by both of these helper T cell subsets, may play an important role in their anti-tumor function [26]. Our previous findings have demonstrated that IL-21-producing CD4^+^ T cell ACT augments the effector CX_3_CR1^+^ CD8^+^ TIL population, resulting in enhanced tumor control [5]. In this study, our data show that BATF, one of the downstream targets of IL-21 signaling, is necessary for CD8^+^ T cell effector differentiation and anti-tumor function. However, it is important to note that BATF expression is not solely dependent on IL-21 signaling and can also be induced by T cell receptor (TCR) stimulation as well as other cytokines such as IL-12 [43,44,45]; thus, our study aims to provide one approach to enhance BATF expression and improve CD8^+^ T cell anti-tumor function.

Recognizing that BATF is an intrinsically necessary transcription factor in CD8^+^ T cell infiltration/survival and function within the tumor, we wanted to test whether BATF overexpression could further enhance CD8^+^ T cell function and tumor control. After observing the significant effects mediated by BATF overexpression, we decided to further investigate potential mechanisms by which BATF regulates CD8^+^ T cells by conducting transcriptional analysis of tumor-infiltrating, retrovirally transduced CD8^+^ T cells. Transcriptomic analyses revealed several effector-like gene signatures, as anticipated. Interestingly, GSEAs also revealed differential enrichment of several gene sets related to the interferon response, hypoxia, cellular stress and senescence, and regulation of rRNA expression (Figure 4 and Appendix A). As demonstrated by a previous publication, REGNASE-1 deficiency reprograms CD8^+^ T cells in the tumor by enhancing BATF, resulting in improving mitochondrial metabolism, which is reversed upon BATF deletion [33]. Similarly, we found a reduction in the hypoxia hallmark gene set (Figure 4A) and oxidative stress-induced senescence gene set (Appendix A) in BATF-overexpressing CD8^+^ T cells, suggesting that these cells are more adapted to the hypoxic tumor microenvironment, resulting in better overall presence and survival in the tumor (Figure 3C). Therefore, further investigation into potential metabolic targets of BATF may provide insight as to how BATF mediates effector CD8^+^ T cell survival and function in cancer. Additionally, as BATF is a member of the Activator protein 1 (AP-1) family of transcription factors [46], it undoubtedly plays a significant role in regulating the transcription of various lineage-defining genes, particularly via cooperative binding between BATF and interferon regulatory factor 4 (IRF4), a transcription factor downstream of T cell receptor signaling [22,47]. Together, BATF and IRF4 function as pioneer transcription factors, remodeling the chromatin landscape [48]. By changing the chromatin landscape, BATF promotes the differentiation of effector CD8 T cells, as observed in the LCMV chronic infection model [22]. However, further investigations into the epigenetic mechanisms by which BATF regulates effector CD8^+^ differentiation in cancer remain of interest.

## 4. Materials and Methods 

### 4.1. Mice

C57BL/6 and C57BL/6 CD45.1/CD45.1 mice were purchased from National Cancer Institute (NCI) (Rockville, MD, USA). C57BL/6 Thy1.1/Thy1.1 Pmel, *Batf ^−/−^*, B6.IL21-VFP knock-in reporter, and *Cd8a^−/−^* mice were obtained from The Jackson Laboratory. Pmel mice were crossed with C57BL/6 mice for at least 4 generations to generate Thy1.2/Thy1.2 Pmel mice. All in vivo experiments used eight- to twelve-week-old male and female mice. All mice used in these studies were bred and maintained under the guidelines approved by the Institutional Animal Care and Use Committee (IACUC) of the Medical College of Wisconsin (AUA00003003).

### 4.2. Bone Marrow Chimeras

Mixed bone marrow chimera (MBMC) mice were generated with *Cd8a^−/−^* recipient mice. *Cd8a^−/−^* mice were irradiated (Gammacell 40 Exactor) with two doses of 500 rad about 4 h apart. Donor bone marrow was obtained from *Cd8a^−/−^* and *Batf^−/−^* mice. Femurs and tibiae were harvested and the bone marrow was flushed using RPMI containing 10% fetal bovine serum (FBS) (Hyclone, Logan, UT, USA). Irradiated mice were intravenously administered 3 × 10^6^ cells consisting of a 30:70 mixture of *Batf^−/−^* and *Cd8a^−/−^* bone marrow cells and allowed eight weeks to engraft. Following the engraftment period, mice were bled to confirm bone marrow reconstitution via flow cytometry prior to experimental use.

### 4.3. Tumor Cell Lines, Inoculation, and Treatments

The B16-F10 tumor cell line was originally purchased from American Type Culture Collection (ATCC) and obtained from Dr. Susan Kaech (Salk Institute, San Diego, CA, USA) in 2013; it has not been further authenticated and tested for *Mycoplasma*. The B16-GP33 tumor cell line was generated as described [49] using the GP33-expressing plasmid generously provided by Hanspeter Pircher (University of Freiburg, Freiburg im Breisgau, Germany). Tumor cells were cultured in DMEM media (Lonza, Morristown, NJ, USA) supplemented with 10% FBS (Hyclone, Logan, UT, USA), 2 mmol/L L-glutamine (Corning, Corning, NY, USA), and 100 U/mL penicillin/streptomycin (Corning, Corning, NY, USA). Tumors were established by subcutaneous injection of 2 × 10^5^ tumor cells into the flanks of mice. Tumor growth was measured using a caliper and tumor volumes were calculated as [length × (width)^2^]/2 (The Jackson Laboratory, Bar Harbor, ME, USA).

### 4.4. Generation of Tumor-Reactive IL-21-Producing CD4^+^ T Cells

Bone marrow cells were isolated from C57BL/6 mice as described above and cultured for one week in RPMI medium (Lonza, Morristown, NJ, USA) containing 10% FBS and 200 ng/mL Flt3L to promote dendritic cell (DC) differentiation. On day 7, DCs were incubated with freeze-thawed tumor lysates at a ratio of three tumor cell equivalents to one DC, as described [50]. After overnight incubation, DCs were maturated with lipopolysaccharide (LPS) (100 ng/mL) for 4–6 h. Mature DCs and purified IL-21-VFP CD4^+^ T cells, isolated from spleen and lymph nodes (Stem Cell Technologies, Vancouver, BC, Canada), were mixed in a 1:5 (DC:T) ratio and cultured under Th17 skewing conditions: IL-6 (100 ng/mL), transforming growth factor beta (TGF-β) (10 ng/mL), IL-23 (20 ng/mL), IL-21 (10 ng/mL), and anti-IL-4 and anti-IFN-γ antibodies (1 μg/mL). Three million IL-21-VFP^+^ activated CD4^+^ T cells were adoptively transferred into recipient tumor-bearing mice inoculated with B16-GP33 melanoma.

### 4.5. Generation of BATF-Overexpressing Tumor-Specific CD8^+^ T Cells

Splenocytes from Pmel Thy1.2/Thy1.2 mice were harvested and mashed against a cell strainer to create a single cell suspension followed by red blood cell lysis (ACK Lysis Buffer, Lonza). Pmel T cells were then activated with gp100 peptide (Genscript, Piscataway, NJ, USA) and 10 IU/mL IL-2 for 24 h, followed by transduction with the MSCV-IRES-Thy1.1 (MIT) retroviral vector (RV) [51]. MIT RV was generously provided by Dr. Susan Kaech. Mouse BATF open reading frame (ORF) (synthesized by Integrated DNA Technologies, IDT) was subcloned into the MIT vector to generate a BATF-expressing vector (MIT-BATF). Following transduction with either empty vector (MIT-Empty) or MIT-BATF, cells were cultured for 24 h. Then, the cells were washed three times to remove residual gp100 peptide and subsequently cultured with 50 IU/mL IL-2, 10 ng/mL IL-7, and 10 ng/mL IL-15. Positively transduced cells (either MIT-Empty or MIT-BATF) were identified by their expression of Thy1.1 surface marker, confirmed by flow cytometry. Five million Thy1.1^+^ positively transduced Pmel CD8^+^ T cells were adoptively transferred by retroorbital injection into each tumor-bearing mouse. CD45.1 mice were used as recipients in order to distinguish adoptively transferred CD8^+^ T cells from endogenous CD8^+^ T cells.

### 4.6. Immune Cell Isolation from Tumors

Tumors were dissected, cut into 2–3-mm sections, and digested in RPMI media containing 10% FBS, 0.7 mg/mL collagenase I (Worthington, Lakewood, NJ, USA), 100 μg/mL bovine pancreatic DNase type I, grade II (Sigma-Aldrich, St. Louis, MO, USA), and 5 mM MgCl_2_ (Sigma-Aldrich) for 45 min at 37 degrees C while shaking. Tumors were then passed through a cell strainer and immune cells were isolated via gradient centrifugation with LymphoPrep (Stem Cell Technologies, Vancouver, BC, Canada, USA). 

### 4.7. Flow Cytometry

Lymphocytes were isolated from tissues as described above. Cells were then stained with antibodies against cell surface antigens for 30–60 min at 4 degrees C. Intracellular and transcription factor staining were performed using the True Nuclear transcription factor buffer set (Biolegend, San Diego, CA, USA). Flow cytometry data were acquired on an LSRII or FACSCelesta (BD Biosciences, San Jose, CA, USA) flow cytometer and analyzed using FlowJo (Treestar, Ashland, OR, USA).

### 4.8. Bulk RNA Sequencing and Analysis

At day 7 post-adoptive transfer, immune cells were isolated from the tumors of B16F10 melanoma-inoculated mice, as described above. For each biological replicate, 2000 positively transduced (Thy1.1^+^) live (7AAD^−^) Pmel T cells from each mouse were directly sorted into lysis buffer containing Protector RNase Inhibitor (Roche, Basel, Switzerland). RNA-seq libraries were prepared using a modified SMART-Seq2 protocol [52] and sequenced with a NextSeq 500 sequencer using a high-output V2 75 cycle kit (Illumina) in 37 bp paired-end mode. Bulk RNA-seq data were aligned to the *Mus musculus* mm10 genome and quality control was performed using the nf-core/rnaseq pipeline v1.4.2 (doi:10.5281/zenodo.1400710, accessed 10 September 2019) [53]. Gene expression was quantified using Salmon [54]. RNA-seq libraries were then normalized and genes were tested for differential expression between MIT-Empty and MIT-BATF samples with DESeq2 v1.24.0 [55]. DESeq2 Wald tests were used to determine whether fold changes were significantly different from zero. For heatmap visualization, data were transformed using the regularized logarithmic transformation [55]. Pre-ranked gene set enrichment analyses (GSEAs) were conducted using shrunken fold-changes and fgsea v1.12.0. The Reactome [56] and Molecular Signatures [57,58] databases were used for GSEA [59]. The Benjamini–Hochberg method was used to adjust *p*-values for false discovery in both differential expression and GSEA analyses [60]. 

### 4.9. Statistical Analysis

Statistical tests for cellular flow cytometry data were performed using GraphPad Prism (version 9.0, GraphPad, San Diego, CA, USA). *p*-values were calculated using two-tailed unpaired Student’s *t*-tests. Kaplan–Meier plot with log-rank (Mantel–Cox) test was used for survival curve analysis. Two-way ANOVA was used for tumor volume curve analysis. * *p* < 0.05 in all data shown.

## 5. Conclusions

Our study reveals the critical role of BATF in maintaining and enhancing functional effector CD8^+^ T cells in response to cancer. BATF overexpression in CD8^+^ T cells enhances CD8^+^ T cell infiltration as well as effector CD8^+^ T cell differentiation and function in a melanoma tumor model. Finally, transcriptomic analyses suggest that BATF promotes a more effector-like and less exhausted gene expression program. Taken together, our findings reveal that BATF overexpression bypasses the need for CD4^+^ T cell help and could be harnessed to provide therapeutic benefits by enhancing effector CD8^+^ T cell survival and anti-tumor functions.

## Figures and Tables

**Figure 1 cancers-13-01263-f001:**
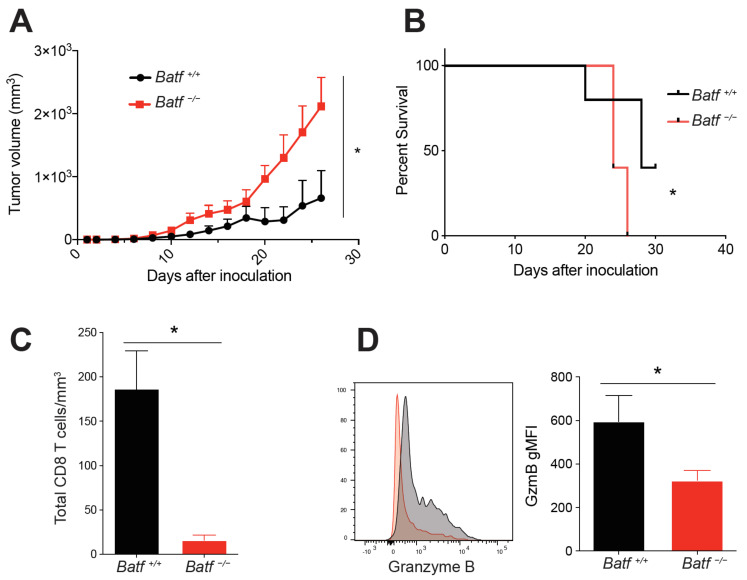
Basic Leucine Zipper ATF-Like Transcription Factor (BATF) is intrinsically required for effector function of tumor-infiltrating CD8^+^ T cells. Mixed bone marrow chimera mice were inoculated with B16 melanoma. Tumor (B16-F10) growth (**A**) and survival (**B**) were monitored over time (*n* = 5). Mixed bone marrow chimera mice were inoculated with B16-GP33 tumors and then sacrificed 8–12 days later, once tumors were palpable. (**C**) Total CD8^+^ T cells/mm^3^ of tumor, assessed via cell counts and flow cytometry (*n* = 4–5 per group). The gating strategy is depicted in Appendix A. (**D**) Granzyme B mean fluorescence intensity (MFI) of activated (CD44^+^) tumor-infiltrating CD8^+^ T cells; representative histograms are shown (*n* = 14 per group). (*, *p*-value < 0.05).

**Figure 2 cancers-13-01263-f002:**
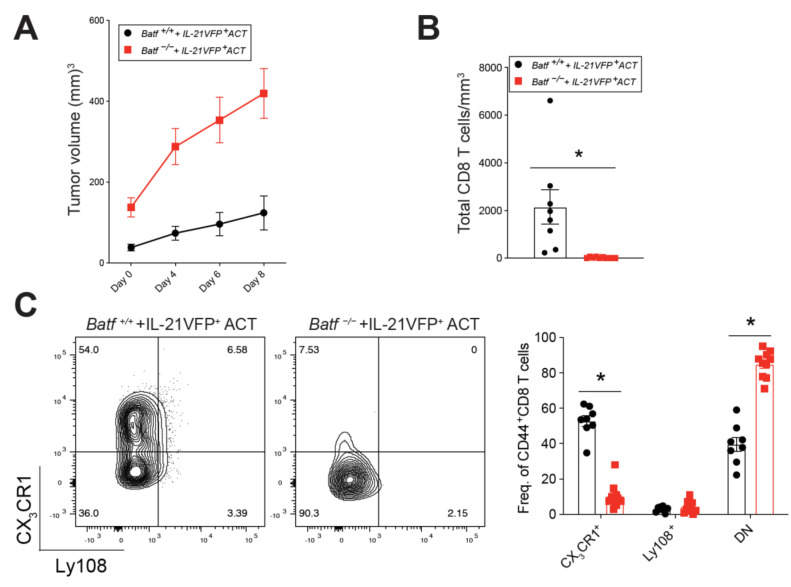
BATF is necessary for endogenous CD8^+^ T cell differentiation in response to adoptive cell transfer (ACT) of interleukin (IL)-21-producing CD4^+^ T cells. Mixed bone marrow chimera mice were inoculated with B16-GP33 melanoma. After tumors were palpable, mice were sub-lethally irradiated and adoptively transferred with IL-21-producing CD4^+^ T cells. Mice were sacrificed after 8 days. (**A**) Tumors were measured to monitor growth following treatment. (**B**) The number of CD8^+^ T cells/mm^3^ of tumor, assessed via cell counts and flow cytometry. (**C**) Frequency of the effector (CX_3_CR1), progenitor (Ly108), and exhausted (double-negative, DN) populations within the activated (CD44^+^) tumor-infiltrating CD8^+^ T cells, along with representative dot plots. Granzyme B (**D**) and PD-1 (**E**) MFI of activated tumor-infiltrating CD8^+^ T cells, with representative histograms shown. (**A**–**E**, *n* = 8–10 mice per group). (* *p*-value < 0.05).

**Figure 3 cancers-13-01263-f003:**
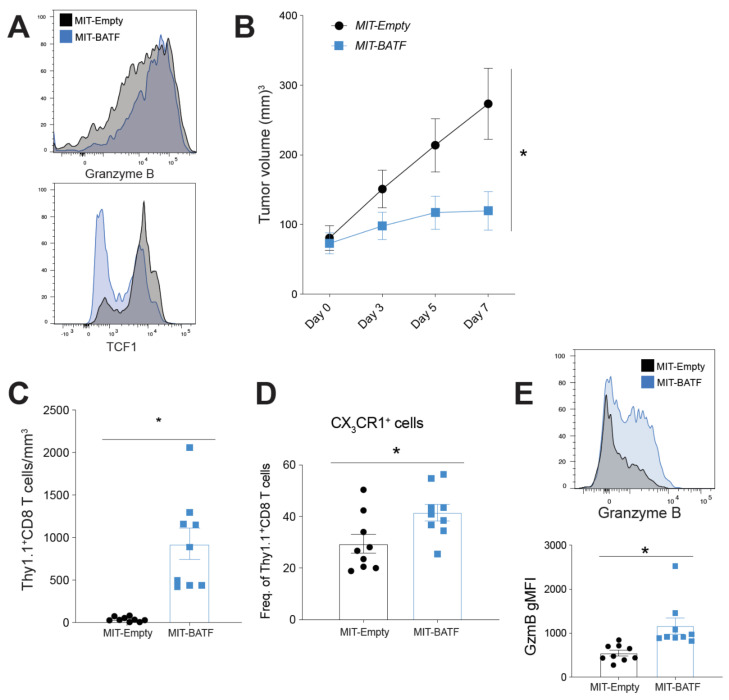
Adoptive transfer of BATF-overexpressing tumor-specific CD8^+^ T cells bypasses the need for CD4^+^ T cell help. Pmel CD8^+^ T cells were activated and transduced with either MIT-Empty- or MIT-BATF-expressing retrovirus. B16-GP33 tumor-bearing CD45.1 mice were then adoptively transferred with MIT-Empty- or MIT-BATF-transduced (Thy1.1^+^), tumor-specific CD8^+^ T cells. (**A**) Granzyme B and TCF1 expression levels of transduced CD8^+^ T cells. (**B**) Tumor volumes were measured following treatment, until sacrifice (*n* = 13 mice per group). (**C**) The number of CD8^+^ T cells/mm^3^ of tumor, assessed via cell counts and flow cytometry. (**D**) Frequency of CX_3_CR1^+^ population within tumor-infiltrating, adoptively transferred transduced CD8^+^ T cells. (**E**) Granzyme B MFI of tumor-infiltrating, adoptively transferred transduced CD8^+^ T cells. (**C**–**E**, *n* = 9 mice per group). (* *p*-value < 0.05).

**Figure 4 cancers-13-01263-f004:**
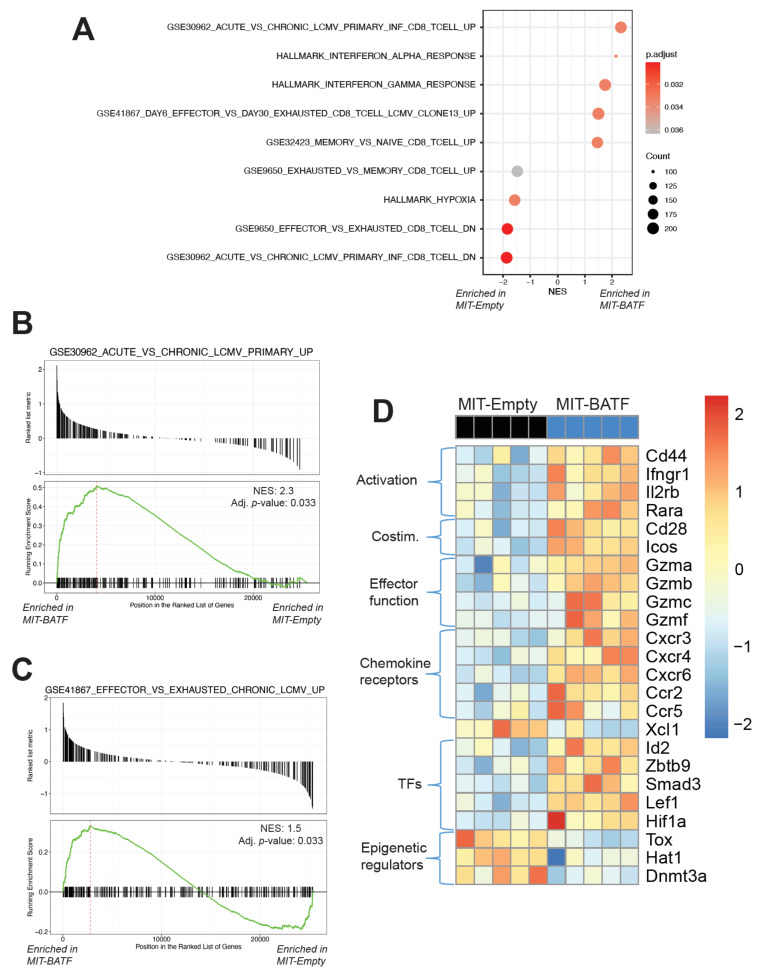
Tumor-infiltrating, adoptively transferred BATF-overexpressing CD8^+^ T cells maintain an effector program via enhanced expression of key genes. B16-F10 tumor-bearing mice were adoptively transferred with MIT-Empty- or MIT-BATF-transduced Pmel CD8^+^ T cells. One week following ACT, tumors were harvested and immune cells were isolated, stained, and florescence-activated cell sorting (FACS)-sorted for Thy1.1^+^ adoptively transferred CD8^+^ T cells. Bulk RNA sequencing was conducted on MIT-Empty and MIT-BATF cells. (**A**–**C**) Gene set enrichment analysis (GSEA) pathways differentially enriched in MIT-Empty and MIT-BATF cells. (**D**) Heatmap of select genes from gene sets and other key markers. (*n* = 5 mice per group).

## Data Availability

The data presented in this study are openly available in GEO, reference number GSE165420.

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
