# Peer review of "Harnessing the IL-21-BATF Pathway in the CD8+ T Cell Anti-Tumor Response"

_cancers, 2021, doi:10.3390/cancers13061263_

Round 1

Reviewer 1 Report

Authors described that BATF is critical to induce effective anti-tumor responses mediated by CD8 T cells. Indeed, data demonstrated that BATF is related to effector CD8 T cells differentiation and function. However, there are critical issues that need to be verified in order to make this conclusion.

Major issues

  1. The authors insisted that CD8 T cell activity is regulated by the IL-21-Barf pathway in anti-tumor responses. To do this, more direct evidences is required and should be confirmed the following: 1) Batf expression or activity in WT CD8 T cells upon TCR or IL-21 stimulation, 2) comparison on proliferation, cytokine productivity and survival of Batf-/- and WT CD8 T cells upon IL-21 or TCR or TCR+IL-21 stimulation.
  2. To address role of Batf in anti-tumor activity of CTL, in vitro cytotoxic assay with activated WT, Batf-/- and BatfTg CD8 T cells should be confirmed.
  3. Although intrinsic effect of Batf could be proven with Batf-/- or Tg CD8 T ACT into tumor bearing animal model, please explain the reason why authors used mixed bone marrow chimeric animal model.
  4. The authors insisted “bypassing the need for CD4 T cell help” in result section 2.3, if this is due to gp100 specific CD8 T cell, it is questionable. Because host has endogenous CD4 T cell population which can give rise to help, so anti-tumor activity of BatfTg or -/- CD8 T cell should be confirmed in CD4 T cell deficient animals.
  5. Expression of IFNg, which is most representative cytokine in CTL function, should be shown with granzyme B level.
  6. The authors used one tumor model (B16 melanoma). To generalize the role of Batf in CD8 T cell anti-tumor responses, it should be confirmed in diverse of tumor model.

Minor issues

  1. Number approved by IACUC should be disclosed.
  2. In figure 4D, the emoticons should be fixed to the numbers.
  3. The author needs to unify the format of all the figures e.g graph label and size.

Reviewer 2 Report

In this manuscript, using a transplanted melanoma model, the authors nicely demonstrated that CD8 T cell-intrinsic BATF expression was required for the differentiation of CX3CR1+ effector T cells and efficient tumor control. Further, the authors convincingly showed that BATF was essential for CD8 T cells to respond to CD4-derived IL-21 help. Importantly, overexpression of BATF can further boost anti-tumor immunity. Together, this manuscript represents a nice continuation of the authors' previous publications. The findings are important to the fields of both T cell biology and tumor immunotherapy. 

Reviewer 3 Report

The manuscript titled “Harnessing the IL-21-BATF pathway in the CD8+ T cell anti-tumor response” is a well-designed research work showing that the expression of Basic Leucine Zipper ATF-Like Transcription Factor (BATF) downstream of IL-21 signaling is critical for maintaining the effector function and anti-tumor cytotoxicity of CD8+ T lymphocytes. It is especially interesting and noteworthy to understand the CD4-CD8 T cell interaction in tumor biology.

However, the manuscript may be improved by addressing the following points:

  •  The reason for using a mixed bone chimera murine model system instead of Cre-LoxP may be addressed (Figure 1 and Figure 2).
  • Similarly, the Thy 1.1 murine model system used in Figure 3 may be described clearly for the benefit of the readers. Why was the model changed?
  • In the Results Section; Line no.88 the authors mention that they detected “very few GP33 tetramer-specific CD8+ T cells”. Since they assessed the total activated CD8+ T cells (Figure 1 and Figure 2) due to this limitation, could these be tumor intrinsic and not infiltrating lymphocytes?
  • The scale and representation of the graphs in Figure 1 and Figure 2 may be improved. The p-values should be mentioned. Please check Fig 2b.
  • In Fig 2A Day 0, is there a significant difference in tumor size?

Reviewer 4 Report

In this manuscript, Topchyan et al demonstrate that BATF is required for the CD8 T cell response to B16-F10 melanoma cells in vivo. They show that chimeric mice in which Cd8a-/- bone marrow was mixed with Batf-/- bone marrow fail to survive subcutaneous injection with B16-F10 tumor cells and make less Granzyme B, PD-1 and CX3CR1 than WT CD8 T cells from these tumors. They show that BATF is required for the increased tumor clearance observed in mice transplanted with IL-21 producing CD4 T cells, consistent with the known induction of Batf mRNA by IL-21 (although that is not shown in this manuscript). Most interestingly, they show that ectopic expression of BATF boosts the CD8 T cell response to B16-F10 by increasing the number of activated effector CD8 T cells. They provide RNA-sequencing data that is consistent with the increased activation of CD8 T cells ectopically expressing BATF. However, no real mechanistic insight is provided into how BATF achieves this activation of CD8 T cells. Overall this was a simple but straight-forward story that demonstrates a role for BATF in the CD8 T cell response to B16-F10, and likely other tumors, and its requirement downstream of IL-21 producing CD4 T cells.

It is important to demonstrate that BATF is required for CD8 T cell activation in tumors, although it is already known to be required in the context of viral infection. However, the strength of this paper lies in its demonstration that BATF is required for generating activated CD8 T cell in response to IL-21 producing CD4 T cells in tumors and that ectopic BATF promotes CD8 T cell activation in the context of a solid tumor. While these two points are demonstrated sufficiently, the manuscript does not resolve whether BATF is required for initial CD8 T cell activation, migration in to tumors, or retention, or survival and does not provide much mechanistic insight.

Specific comments:

In multiple places the authors overstate the conclusions that can be drawn from the data. For example:

  1. In the abstract, “Mechanistically, transcriptomic analyses revealed that BATF enhanced CD8+ T 27 cell activation and function by increasing the expression of costimulatory receptors, effector molecules, and transcriptional regulators”. They do not provide any real mechanistic insight in this paper and certainly they do not show that BATF functions by increasing the expression of costimulatory receptors. They only show decreased PD-1 expression on CD8 T cells, and as an inhibitory receptor expressed on activated CD8 T cells PD-1 is expected to be low since Batf is required for CD8 T cell activation, as also indicated by decreased CX3Cr1.
  2. In Figure legend 1 title: “BATF is intrinsically required to maintain tumor-infiltrating CD8+ T cell effector function” But their data does not show that BATF maintains CD8 T cells rather than being required for migration or some other effect. They can only say that BATF is required for the generation of activated CD8 T cells in the context of B16-F10.
  3. Similarly, on line 149 they say “BATF overexpression downregulated the transcription factor TCF1” when there are shifts in the population with TCF1 expressing naïve or exhausted cells being reduced among MIT-BATF expressing cells. This could well be a very indirect effect as seems to be indicated by the overall activation phenotype of MIT-BATF expressing cells.
  4. They also claim that the chimeras prove that BATF is required in CD8 T cells but it is still feasible that BATF was required in less mature cells to establish competent naïve CD8 T cells. This caveat should be mentioned.

The gene expression profiles provided for the MIT-BATF expressing CD8 T cells are consistent with the proposed activation of CD8 T cells. However,

  1. Somewhere in the paper the authors should discuss the mechanisms by which BATF could function. For example, is BATF function alone or in collaboration with IRF4/8? Are any of these genes known IRF4/8 targets? Are any even known targets of BATF or AP-1?
  2. Can the authors show how much over expression of BATF is occurring either at the mRNA level, or preferably, at the protein level?

Additional issues:

  1. Some figure legend titles are bolded and some are not. Figure lettering is also confusing.
  2. Which Batf-/- mice are being used? There is no reference to these mice anywhere in the manuscript.
  3. How was the MIT-BATF vector generated? Is this human or mouse?
  4. They mention looking at CD44hi activated CD8 T cells but the data presented in Figure 1 are so heavily processed that I am not sure whether the numbers in 1C are total CD8 or CD44hi CD8? Please clarify, and maybe show a flow cytometry plot of the gating strategy.
  5. Why are the letters for each section of a figure under the figure?

Round 2

Reviewer 1 Report

Although the authors have re-written a few parts of the manuscript, they have not fully addressed the major concerns raised in the initial review.

In regard to ACT model, I didn’t suggest and ask that authors use BATF null mutant model for anti-tumor response. The most general animal model for CD8 T cell-mediated anti-tumor response is adaptive cell transfer (ACT) in which CD8 T cells are isolated and in vitro stimulated and transferred into tumor bearing mice [lots of reference]. My issue was why authors used BM chimera model instead of using this simple and powerful model that can directly prove the role of BATF in CD8 T cell-anti-tumor responses. The experiment using conventional ACT model is required and should be demonstrated, since it could be linked to CD4 T cell help in which is mainly mediated by IL-21.

In regard to CD4 T cell help, authors argue and predict that endogenous CD4 T cells would not have help function, because they are suppressed by immunosuppressive TME or Treg cells are dominant in tumor bearing mice. As authors have shown in Fig1A, +/+ BMC mice controlled tumor growth better than -/- BMC mice, I don’t understand the result without endogenous CD4 T cell help, because BATF is induced by IL-21 which mainly produced by CD4 T cells and involved in efficient generation of CD8 effector T cells. If it not the case, please discuss how +/+ BMC is better and which cells produce IL-21 in +/+ BMC mice. I knew very well that you have shown in Cell Reports that CD4 T help is critical for effector CD8 T cell in chronic LCMV infection model using CD4 depletion. Current research is not using the LCMV model, but the tumor model. Like your previous publication, authors should confirm it with the relevant evidence, but not with prediction and reference.

In regard to tumor model, I fully understand melanoma model specially in the field of immunology and also understand what model authors used in this study. As authors know, in particular, since the generalization process is very important in cancer research, it is difficult to accept the results of this study as a reviewer if you refuse to publish the relevant data.        

Reviewer 3 Report

The authors fairly addressed my previous concerns. However, the authors need to present the gating strategies (FACS) in the figures. In particular, illustrate the populations of Thy1.1+ and Thy1.1- (or Thy1.2+) cells. In addition, in Appendix Fig. 1, the gating methods are not appropriate.

Reviewer 4 Report

The authors have largely addressed my comments by modifying the manuscript. I only have two minor comments as indicated below.

The authors changed the sentence “Mechanistically, transcriptomic analyses revealed that BATF enhanced CD8+ T 27 cell activation and function by increasing the expression of costimulatory receptors, effector molecules, and transcriptional regulators” to “Transcriptomic analyses revealed that BATF overexpression in CD8+ T cells increased expression of costimulatory receptors, effector molecules, and transcriptional regulators, which may contribute to their enhanced activation and effector function”. This sentence still implies that these changes are directly due to BATF rather than due to increased activation. How about this: “Transcriptomic analysis revealed that BATF overexpressing CD8+ T cells had increased expression of…….”

Similarly, “BATF overexpressing cells showed reduced expression of on downregulated the transcription factor TCF1” needs to be changed to “BATF overexpressing cells showed reduced expression of the transcription factor TCF1.”

Round 3

Reviewer 1 Report

I still couldn’t find any reason why authors can't do the conventional ACT experiment (optimal method for endogenous BATF role), nor have I been convinced of authors’ response. No additional experiments have been carried out and included.

No clear verification was made through further CD4 depletion experiment.

Results without generalization provide very limited information and cause doubts about the results. Correcting the title doesn't seem to solve the problem.

Reviewer 3 Report

The authors fairly addressed my preview concerns.